# Highly Efficient Adsorption of Tetracycline Using Chitosan-Based Magnetic Adsorbent

**DOI:** 10.3390/polym14224854

**Published:** 2022-11-11

**Authors:** Franciele da Silva Bruckmann, Carlos Eduardo Schnorr, Theodoro da Rosa Salles, Franciane Batista Nunes, Luiza Baumann, Edson Irineu Müller, Luis F. O. Silva, Guilherme L. Dotto, Cristiano Rodrigo Bohn Rhoden

**Affiliations:** 1Programa de Pós-Graduação em Química, Universidade Federal de Santa Maria—UFSM, Santa Maria 97105-900, Brazil; 2Research Group on Adsorptive and Catalytic Process Engineering (ENGEPAC), Universidade Federal de Santa Maria—UFSM, Santa Maria 97105-900, Brazil; 3Universidad de la Costa, CUC, Calle 58 # 55–66, Barranquilla 080002, Colombia; 4Laboratório de Materiais Magnéticos Nanoestruturados—LaMMaN, Universidade Franciscana—UFN, Santa Maria 97010-030, Brazil; 5Programa de Pós-Graduação em Nanociências, Universidade Franciscana—UFN, Santa Maria 97010-030, Brazil

**Keywords:** antibiotics, emerging pollutants, iron oxide nanoparticles, magnetic nanocomposites

## Abstract

Herein, tetracycline adsorption employing magnetic chitosan (CS·Fe_3_O_4_) as the adsorbent is reported. The magnetic adsorbent was synthesized by the co-precipitation method and characterized through FTIR, XRD, SEM, and VSM analyses. The experimental data showed that the highest maximum adsorption capacity was reached at pH 7.0 (211.21 mg g^−1^). The efficiency of the magnetic adsorbent in tetracycline removal was dependent on the pH, initial concentration of adsorbate, and the adsorbent dosage. Additionally, the ionic strength showed a significant effect on the process. The equilibrium and kinetics studies demonstrate that Sips and Elovich models showed the best adjustment for experimental data, suggesting that the adsorption occurs in a heterogeneous surface and predominantly by chemical mechanisms. The experimental results suggest that tetracycline adsorption is mainly governed by the hydrogen bonds and cation–π interactions due to its pH dependence as well as the enhancement in the removal efficiency with the magnetite incorporation on the chitosan surface, respectively. Thermodynamic parameters indicate a spontaneous and exothermic process. Finally, magnetic chitosan proves to be efficient in TC removal even after several adsorption/desorption cycles.

## 1. Introduction

The intense growth of the pharmaceutical industry and the large consumption of drugs worldwide have contributed to water contamination and the environment. Antibiotics are a class of drugs widely used to control infectious diseases that affect humans and animals [1]. Among them, tetracycline (TC) is one of the most used drugs due to its broad spectrum of action and cost-benefit. However, tetracycline is excreted from the body in high concentrations in the unchanged form [2]. Furthermore, considering its physicochemical characteristics, such as high-water solubility, structural complexity, and low degradability, TC has been widely found in aquatic environments with concentrations in the range of ng L^−1^ [3].

Antibiotics are included in the class of emerging pollutants due to their ability to alter the balance of the microbiota and develop bacterial resistance to multiple drugs. Additionally, studies have reported various side effects caused by tetracycline on human health, such as thyroid dysfunction, liver toxicity, and effects on bone growth [3,4].

In this scenario, several environmental remediation techniques to solve the problem of water contamination by tetracycline have been explored, such as photocatalysts [5,6], Fenton [7], ozonation [8], and adsorption using nanomaterial-based systems [9]. Among them, the adsorption is a tertiary method widely used due to its easy operation, without by-products’ generation, and high efficiency in pollutant removal at very low concentrations (milligrams and nanograms per liter) [10]. Nonetheless, the adsorption efficiency is not only dependent on adsorbent characteristics but is also involved with the affinity by adsorbate molecules [11,12].

Regarding the intrinsic characteristics of the adsorbent, an important point regarding the performance of the process is related to its textural properties and physicochemical stability. Thus, nanotechnology is a valuable tool for the design and synthesis of multifunctional materials with promising applications [13,14].

A large variety of nanoadsorbents have been used in emerging pollutant removal, such as carbon nanomaterials, magnetic nanoparticles, and polymeric and siliceous compounds [14,15,16,17,18,19]. Among these adsorbent materials, chitosan is largely explored in adsorption studies considering its notable characteristics, such as low toxicity, biodegradability, abundance of functional groups, and its important feasibility for interaction with binding sites. Although CS exhibits remarkable advantages, the sensitivity to acidic pH enables the solubilization of polymeric chains, which decreases its efficiency as an adsorbent [20].

Thus, the synthesis of chitosan-based materials is a strategy to overcome the drawbacks as well as enhance the adsorbent performance. Chitosan is used to stabilize magnetic nanoparticles to decrease surface energy, which is related to the tendency of agglomeration and cohesion [21,22].

Furthermore, the improvement of the physicochemical stability of both compounds and the magnetic behavior allows the easy separation of the adsorbent material from the aqueous solution. Additionally, chitosan-based magnetic derivatives show a higher specific surface area than pure polymer [22,23].

In this work, a simple and direct approach was used to synthesize a chitosan-based magnetic adsorbent (CS·Fe_3_O_4_) under mild conditions, controlling the amount of iron oxide nanoparticles on the polymer surface employing only one iron specie (FeCl_2_). Furthermore, the effect of different experimental conditions on the TC adsorption behavior was investigated and discussed. Isotherm and kinetics modeling, as well as thermodynamic parameters for the adsorption of tetracycline on CS·Fe_3_O_4_, were also determined.

## 2. Materials and Methods

### 2.1. Synthesis of Magnetic Chitosan

The chitosan-based magnetic adsorbent was synthesized by the co-precipitation method employing only iron (II) chloride tetrahydrate (FeCl_2_·4H_2_O) as the iron salt. The precipitation of iron ions was performed by adding ammonium hydroxide until reaching pH 10.0. For more details about the magnetization procedure, see previous works by our research group [21,22]. The scheme in Figure 1 summarizes the straightforward obtaining of magnetic chitosan using only Fe^2+^ ions under mild conditions.

### 2.2. Adsorbent Characterization

The starting material (chitosan) and magnetic adsorbent (CS·Fe_3_O_4_) were characterized by different techniques. The functional groups and crystalline patterns were determined through Fourier transform spectroscopy (Perkin–Elmer, model Spectro One, São Paulo, Brazil) and X-ray diffraction (Bruker diffractometer, model D2 Phaser, Billerica, MA, USA), respectively. The morphological characteristics and magnetic properties were evaluated using a scanning electron microscope (Zeeis Sigma 300 VP, Sigma-Aldrich, St. Louis, MO, USA) and vibrating sample magnetometer (Starford Research Systems Model SR830 DSP lock-in amplifier coupled to a Starford Research Systems low-noise pre-amplifier model SR560, including a Kepco bipolar operation power supply). Textural properties were analyzed by nitrogen adsorption–desorption isotherms (S_BET_ and pore size) using the Brunauer–Emmett–Teller (BET) and Barret–Joyner–Halenda (BJH) equations (ASAP 2020 Plus Micromeritics equipment). Meanwhile, the surface charge was determined by the 11-point experiment (zero charge of potential, pH_ZCP_).

### 2.3. Adsorption Procedure and Mathematical Modeling

The tetracycline adsorption was performed in batch experiments. For comparative purposes, chitosan and magnetic chitosan were initially used in an adsorption test to determine the influence of magnetite incorporated on the biopolymer surface. The adsorption equilibrium study was performed using CS·Fe_3_O_4_ (0.5 g L^−1^), adsorbate concentration 50 mg L^−1^, pH = 7.0, at different temperatures (20, 30, and 40 °C). The adsorption kinetic was evaluated varying the TC concentrations (10–200 mg L^−1^), maintaining the temperature constant (20 °C). The thermodynamic study was estimated using the same conditions as the equilibrium study. Meanwhile, the influence of pH was verified in a pH range between 4 and 10.0, using HCl and NaOH solutions (0.1 mol L^−1^) for adjusting the pH medium. The influence of ionic strength was investigated employing a gradual concentration of sodium chloride (0.01–1.0 mol L^−1^). The regeneration of the adsorbent was performed using NaOH as the desorbing agent (0.25 mol L^−1^) at room temperature for 1 h (120 rpm). The residual concentration of tetracycline in solution was measured in a UV-vis spectrophotometer (Shimadzu, São Paulo, Brazil) at λ = 380 nm. The removal percentage and adsorption capacity at equilibrium were calculated by the following equations [23]:(1)$$R\%=\frac{C_{0}-C_{e}}{C_{0}}100$$
(2)$$q_{e}=\frac{\left(C_{0}-C_{e}\right)\;V}{m}$$
where *C*_0_ is the initial concentration of TC (mg L^−1^), *C_e_* is the concentration of TC at equilibrium (mg L^−1^), *q_e_* is the adsorption capacity at equilibrium, and *m* is the mass of CS·Fe_3_O_4_ (g).

Adsorption isotherm models are useful to understand the relationship between the adsorbate and adsorbent material, predict the adsorption capacity, and find the effect of temperature on the adsorption equilibrium. The Langmuir, Freundlich, and Sips isotherms were used to fit the experimental data. Non-linear expressions of these models are shown below [13,22]:(3)$$q_{e}=\frac{q_{max}\;\;K_{L}\cdot C_{e}}{1+K_{L\;}C_{e}}$$
(4)$$q_{e}=K_{F}\;C_{e}^{1/n}$$
(5)$$q_{e}=\frac{q_{s}\left(K_{s}C_{e}^{ns}\right)}{1+(K_{s}C_{e}^{ns})}$$
where *q_max_* is the maximum adsorption capacity (mg g^−1^), *K_L_* is the Langmuir constant (L mg^−1^), *K_F_* is the Freundlich constant ((mg g^−1^) (mg L^−1^)^−1/n^), *K_s_* is the Sips constant (L mg^−1^), *n* is the Freundlich constant related to intensity, and *ns* is the Sips constant associated with the heterogeneity.

Kinetic models are important tools that help to understand the mass transfer phenomenon, the effect of contact time on the adsorption rate, as well as to describe the possible mechanisms involved in the adsorption process. Pseudo-first-order (PFO), pseudo-second-order (PSO), and Elovich models were used to construct the kinetic profile and determine the adjustment parameters for TC adsorption onto magnetic chitosan. The mathematical expressions corresponding to the kinetic models are shown in Equations (6)–(8), respectively [15]:(6)$$q_{t}=q_{1}\left(1-e^{-k_{1}\cdot t}\right)$$
(7)$$q_{t}=\frac{t}{\left(\frac{1}{k_{2}\;\;q_{2}^{2}}\right)+\left(\frac{t}{q_{2}}\right)}$$
(8)$$q_{t}=\frac{1}{\beta }ln\left(\alpha \beta t\right)$$
where *q_t_* is the quantity of TC adsorbed at time *t* (mg g^−1^), *q*_1_ and *q*_2_ are the theoretical adsorption capacities (mg g^−1^) of PFO and PSO models, respectively, *k*_1_ (min^−1^) and *k*_2_ (g mg^−1^ min^−1^) are the rate constants of pseudo-first-order and pseudo-second-order kinetic models, respectively, *α* is the constant that describes the initial sorption rate (mg g^−1^ min^−1^), and *β* is the constant related to the degree of surface coverage and activated energy (g mg^−1^).

Thermodynamic parameters (∆*G*^0^, ∆*H*^0^, and ∆*S*^0^) were calculated by the Van ’t Hoff equation to determine the effect of temperature on the adsorption behavior. Thermodynamic calculations are presented in Equations (9)–(11) [24]:(9)$$∆G=-RT\;lnK_{c\;}$$
(10)$$∆G^{0}=∆H^{0}-T∆S^{0}$$
(11)$$ln\left(K_{C}\right)=\frac{∆S^{0}}{R}-\frac{∆H}{RT}$$
where *K_c_* is the thermodynamic constant, *R* is the universal gas constant (kJ mol^−1^ K^−1^), and *T* is the absolute temperature (K).

The equilibrium and kinetic data were estimated through non-linear regression using the Statistical 10 software (StatSoft, Tulsa, OK, USA). To verify the validity of mathematical models, different error functions were used. The corresponding equations of the coefficient of determination (*R*^2^), adjusted coefficient of determination (*R*^2^*_adj_*), sum of squared errors (*SSE*), and average relative error (*ARE*) are expressed as follows [22]:(12)$$R^{2}=1-\frac{\sum_{i=1}^{n}{\left(q_{e}{}_{,exp}-q_{e,}{}_{pred}\right)}^{2}}{\sum_{i=1}^{n}{\left(q_{e,}{}_{exp}-{\bar{q_{e,}}}_{exp}\right)}^{2}}$$
(13)$$R_{adj}^{2}=1-\left[\frac{\left(1-R^{2}\right)\left(n-1\right)}{n-k-1}\right]$$
(14)$$SSE=\frac{1}{n}\sum_{i=1}^{n}{\left(q_{e,\;exp}-q_{e,\;pred}\right)}^{2}$$
(15)$$ARE=\frac{100}{n}\sum_{i=1}^{n}\left|\frac{q_{e,\;exp}-q_{e,\;pred}}{q_{e,\;exp}}\right|$$
where *q_e,exp_* is the adsorbed quantity, obtained experimentally, *q_e,pred_* is the adsorption capacity predicted by isotherm and kinetic models, *n* is the number of data points, and *k* is the number of parameters in the isotherm and kinetic models.

## 3. Results and Discussion

### 3.1. Fourier Transform Infrared Spectroscopy (FTIR)

Figure 2 shows the FTIR spectra of chitosan and magnetic chitosan. As shown in the chitosan spectrum, the characteristic bands around 3400 cm^−1^ are related to the stretching vibration of amine and hydroxyl groups (NH_2_ and OH). Peaks at 2927 and 2846 cm^−1^ can be attributed to C–H stretching, while bands in the region around 1636–1612 cm^−1^ are assigned to in-plane bending vibrations (NH_2_). The remaining peaks (1380 and 1071 cm^−1^) refer to CH_3_ symmetric stretching and C–O stretching, respectively [22,25]. Regarding the FTIR spectrum of magnetic chitosan, a slight shift of the band from 1615 to 1620 cm^−1^ related to in-plane bending vibrations and displacement from 1382 to 1385 cm^−1^ attributed to CH_3_ symmetric stretching, as well the appearance of a new peak at 615 cm^−1^, suggest the chitosan coating with magnetite nanoparticles [26,27].

### 3.2. X-ray Diffraction (XRD)

Crystalline structures of chitosan and magnetic chitosan were determined by XRD (Figure 3a,b). According to the diffractogram of pure chitosan (Figure 3a), two characteristic peaks can be observed at 2θ ≈ 10° (020) and 19° (110), which are attributed to crystalline patterns (crystals I and II, respectively) [28]. For the magnetic adsorbent (Figure 3b), five typical peaks were verified at 2θ ≈ 30° (220), 35° (311), 40° (400), 57° (511), and 62° (440). These XRD patterns are characteristic of magnetite with a spinel structure [24,27]. Furthermore, a significant decrease in the intensity of the main crystal of the biopolymer at 2θ ≈ 19° (110) can be assigned to iron oxide nanoparticle functionalization with chitosan [22].

### 3.3. Scanning Electron Microscopy (SEM)

Morphological features of chitosan and magnetic chitosan are shown in Figure 4a,b. As shown in the SEM image (Figure 4a), the unmodified chitosan presented an irregular and rough surface. Hao et al. [29] reported that the chitin deacetylation temperature and material crystallinity can affect morphology. SEM micrographs of CS·Fe_3_O_4_ clearly show the deposition of iron oxide onto the polymer surface. Additionally, magnetic nanoparticles are distributed homogeneously and coat the polymeric matrix due to the high proportion of iron precursor for chitosan used in the synthesis (10:1, mass:mass) [21].

### 3.4. Vibrating Sample Magnetometer (VSM)

Figure 5 shows the magnetization curve of CS·Fe_3_O_4_ at room temperature. According to hysteresis loops, the chitosan-based magnetic adsorbent exhibited a ferromagnetic behavior with a magnetization saturation value of 43 emu g^−1^. Similar results were reported by Prabha and Raj [30] and Ates et al. [31] when synthesizing composites of Fe_3_O_4_-chitosan particles through the classical co-precipitation method (magnetic nanoparticles showed MS values of 40 and 37 emu g^−1^, respectively).

### 3.5. Nitrogen Porosimetry

The N_2_ adsorption–desorption isotherm plots of chitosan and magnetic chitosan were used to determine the specific surface area by the Brunauer–Emmett–Teller (BET) method. Meanwhile, the Barrett–Joyner–Halenda (BJH) equation was employed to calculate the pore size distribution. According to Figure 6a, it was possible to verify that the specific surface area increased with the magnetite incorporation into the chitosan structure. Thus, the pure biopolymer and magnetic nanocomposite exhibited S_BET_ values of 0.59 and 47.58 m^2^ g^−1^, respectively. Additionally, according to the IUPAC classification, the isotherms present an H3-type hysteresis, which is typical of mesoporous materials [22].

Regarding the pore size distribution, a significant decrease in this parameter was caused by the chemical modification (i.e., the average pore size reduced from 49.78 to 12.60 nm). Similar results were reported by Pylypchuk and coauthors [32] when synthesizing chitosan/magnetite nanocomposites through the co-precipitation method. On the other hand, the pore volume increased with the incorporation of magnetite, which can be attributed to intramolecular interactions between chitosan molecules [33].

### 3.6. Tetracycline Adsorption

#### 3.6.1. Effect of Magnetite Incorporation onto Chitosan Surface

For comparative purposes, a study was performed employing the starting material (chitosan) used in the magnetic adsorbent synthesis. The influence of magnetite-functionalization chitosan on the adsorption capacity and removal percentage is shown in Figure 7. From the results, it was possible to verify that the iron oxide nanoparticles had a significant effect on the adsorption efficiency. For instance, non-magnetic material exhibited maximum adsorption capacity and removal values of 50.86 mg ^−1^ and 49.74%, respectively. On the other hand, the chitosan-based magnetic adsorbent displayed higher values for *q_max_* and removal (74.71 mg g^−1^ and 73.69%, respectively). Previous studies of our research group reported the effect of the textural properties of chitosan and its derivatives on the adsorption efficiency. Chitosan-based magnetic adsorbents presented greater specific surface areas than the non-modified biopolymer, as well as a lower pore size [21,22].

#### 3.6.2. Effect of Initial Concentration of TC and Adsorbent Dosage

To investigate the effect of adsorbate on the adsorption behavior, different concentrations of tetracycline were used (10, 25, 50, 100, and 200 mg L^−1^). Figure 8 shows the influence of the initial concentration of TC. According to the graph, it was possible to observe that the removal percentage and the maximum adsorption capacity (*q_max_*) were highly dependent on the TC concentration. The *q_max_* values increased proportionally with the concentration, and the maximum adsorption was reached with 200 mg L^−1^ (141.21 mg g^−1^). On the other hand, the removal percentage had a significant enhancement when the concentration was increased from 10 to 50 mg L^−1^ (e.g., the percentage value changed from 52.72% to 73.69% when the concentration of TC was increased from 10 to 50 mg L^−1^).

These results are explained by the equilibrium between the adsorption sites and the TC molecules (number of sites:molecules adsorbed). With the gradual increase of the adsorbate molecules, the saturation of the adsorption sites occurred more quickly. Similar behaviors were reported by Da Silva Bruckmann et al. [2] and Nasiri et al. [1] when removing tetracycline using GO·Fe_3_O_4_ and CuCoFe_2_O_4_@Chitosan.

Another important experimental parameter that can affect the adsorption efficiency is the adsorbent dosage. In this study, five different concentrations (0.125–1.0 g L^−1^) were used to determine the optimal adsorption condition. As demonstrated in Figure 9, the removal percentage increased with the CS·Fe_3_O_4_ dosage (i.e., raising the concentration from 0.125 to 0.75 g L^−1^, the removal value increased from 52.7% to 76.33%). Although the removal efficiency improved proportionally to dosage, at the highest concentration (1.0 g L^−1^), a decrease in the removal value was observed (≈25%). The reason for this behavior can be related to the occurrence of adsorbent–adsorbent interactions, reducing the specific surface area to interact with adsorbate molecules [23].

The adsorbent dosage has an important influence on the adsorption behavior, considering the effect of the surface and contact areas, i.e., with an increase in the amount of adsorbent, more binding sites are available for interaction, and consequently, a high quantity of TC is adsorbed [13]. However, there was a significant reduction in the adsorption capacity values with the increase in the amount of adsorbent tested. These findings can be attributed to the constant volume and concentration of tetracycline. Thus, most adsorption sites remained unoccupied, which justifies the considerable decrease in *q_max_* values [1].

#### 3.6.3. Effect of pH and Adsorption Mechanisms

The pH exerts an important role in adsorption performance due to its effect on the chemical speciation of the adsorbate and the surface charge of the adsorbent material [34]. In this adsorption batch experiment, the influence of the pH on TC removal was evaluated within the pH range 4–10. Figure 10a,b show the effect of pH on the zero potential of charge (pH_ZPC_) and TC adsorption. As shown in Figure 10b, it was possible to observe that the adsorption of tetracycline on CS·Fe_3_O_4_ was significantly affected by the pH of the solution.

The maximum adsorption capacity and removal percentage were reached at pH 7.0 (74.71 mg g^−1^ and 73.69%, respectively). Tetracycline is a polyprotic molecule with three ionization constants (pK_a_ 3.3, 7.3, and 9.4) [35]. Thus, functional groups can be easily protonated and deprotonated depending on the acidity or alkalinity of the aqueous medium. Under acidic conditions, both compounds (adsorbate and adsorbent) had a positive charge, decreasing the affinity of TC for adsorption sites due to competition with H^+^ ions [14]. On the other hand, with increasing pH, the cationic species of the pollutant (TCH_3_^+^) and CS·Fe_3_O_4_ were attenuated, improving the adsorption efficiency [22]. In neutral medium, tetracycline acquires a zwitterionic character (TCH_2_^0^), interacting easily with the uncharged adsorbent surface (pH_ZPC_ = 7.46). Therefore, the occurrence of hydrogen interactions is favored [2,34].

With the gradual increase of pH, a significant decrease in the adsorption efficiency could be observed (for instance, the removal value reduced from 73.69% to 25.97% when the pH was raised from 7.0 to 10). In an alkaline environment, the adsorbent and adsorbate exhibit anionic characteristics, hampering the mass transfer from the liquid to the solid phase [36].

The adsorption mechanisms are quite specific and involve the pH and chemical structure of the adsorbent and the adsorbate. In this study, some hypotheses about the adsorption of TC on CS·Fe_3_O_4_ were proposed (Figure 11). Considering the hydrophilic character of chitosan and the characteristic structure of tetracycline, the adsorption phenomenon is involved with hydrogen interactions (dipole–dipole hydrogen bonding and Yoshida H-bonding) [22]. Corroborating with this mechanistic hypothesis, the sensitivity of adsorption to pH variations also indicates that the chemical species of TC and adsorbent surface charge are affected by changes in the pH range (acidity and alkalinity), altering their electrostatic interactions [14]. Despite the hydrogen bonds having an important contribution to the TC adsorption, the improvement in the removal percentage and maximum adsorption capacity with the incorporation of magnetite into the chitosan matrix suggests that the cation–π interactions are mainly involved in the process [23,37].

#### 3.6.4. Effect of Ionic Strength

Due to the presence of different inorganic ions in wastewater treatment systems, the effect of ionic strength on TC adsorption performance was investigated using sodium chloride at different concentrations (0.01–1.0 mol L^−1^) under pH = 7.0, with constant adsorbate and adsorbent concentrations (50 mg L^−1^ and 0.5 g L^−1^, respectively). According to Figure 12, it was possible to verify a considerable decrease in the maximum adsorption capacity values at higher concentrations of NaCl. For instance, the *q_max_* value reduced from 70.21 to 14.54 mg g^−1^ with the increase in the molarity from 0.01 to 1.0 mol L^−1^. The ionic strength can affect not only the interactions between the adsorbate and the adsorbent but also decrease the aqueous solubility of the pollutant [38,39]. The reduction in adsorption capacity can be attributed to the binding of Na^+^ and Cl^−^ ions to the zwitterionic form of TC, hindering its interaction with the adsorption sites [40].

#### 3.6.5. Kinetic Modeling

The mass transfer phenomenon and the control of the tetracycline adsorption rate were determined using pseudo-first-order (PFO), pseudo-second-order (PSO), and Elovich kinetic models. The kinetic study of TC adsorption onto CS·Fe_3_O_4_ was carried out with different initial concentrations of the adsorbate (10–200 mg L^−1^) at room temperature. Table 1 shows the non-linear estimation of kinetic adjustment parameters for TC adsorption.

From the results presented in Table 1, it was possible to verify that the PFO and PSO models were not suitable to describe the experimental data at higher concentrations due to the lowest determination coefficient (*R*^2^) and highest error function values compared to the Elovich model. On the other hand, at concentrations of 10 and 25 mg L^−1^, the three kinetic models were adequate to adjust the kinetic data [15,41]. Thus, the Elovich model demonstrated the best adjustment for the TC adsorption considering the high *R*^2^ values (*R*^2^ ≥ 0.993) and low values for ARE and SSE. This kinetic model assumes heterogeneous surface adsorption with the occurrence of chemisorption mechanisms. Additionally, neither the desorption phenomenon nor interactions between the adsorbed solutes can significantly affect the adsorption kinetics at low surface coverage [42].

Regarding the constant related to the sorption rate (α), it can be observed that the α values increased with the initial concentration of TC, suggesting that the adsorption phenomenon involves chemical interactions. At the same time, the substantial decrease verified in the constant refers to the surface coverage (β) together with the increases in the TC concentration, indicating an adsorption process with different energy levels (heterogeneous adsorption sites) [43].

Figure 13 displays the kinetic curves for TC adsorption onto CS·Fe_3_O_4_ at different concentrations obtained by the Elovich model. The kinetic profile for TC adsorption revealed a typical kinetic profile with different stages. Initially, a high amount was adsorbed, followed by slow steps until equilibrium was achieved. It is noticeable that the adsorption capacity improved with the increase of the initial concentration of tetracycline. The removal percentage within 3 h of contact time varied between 34.99% and 73.69%. These results can be attributed to the more efficient occupation of vacancy sites at the highest adsorbate concentrations [44].

#### 3.6.6. Adsorption Equilibrium Isotherms and Thermodynamic Study

Isotherm models are important tools for understanding the adsorption equilibrium behavior at different temperatures and adsorbate/adsorbent interactions. In this study, three models, Langmuir, Freundlich, and Sips, were used for experimental data adjustment. The equilibrium parameters for TC adsorption onto magnetic chitosan are shown in Table 2.

According to the results presented in Table 2, it was possible to verify that the Sips isotherm represented the best adjustment for adsorption equilibrium data due to high *R*^2^ values (*R*^2^ ≥ 0.987) and low values for error functions (ARE and SSE). Sips is a model derived from the Langmuir and Freundlich expressions used to overcome the limitations of these models [45]. Additionally, ns values greater than 1 suggest positive cooperativity, and therefore, the lateral involvement of the adsorbate molecules [46]. In addition, it is noticeable that the adsorption was affected by the temperature, i.e., the increase in the temperature caused a significant decrease in the maximum adsorption capacity. The considerable decrease in the *q_max_* values with the temperature rise suggests the occurrence of exothermic adsorption. Additionally, the temperature can alter the adsorbate solubility and modify the adsorbent structure, which can cause deactivation and destruction of active sites [10,47].

The effect of temperature on the TC adsorption behavior on magnetic chitosan was investigated under different experimental conditions (293.15, 313.15, and 333.15 K). The thermodynamic parameters (Gibbs free energy variation values (Δ*G*^0^), enthalpy (Δ*H*^0^), and entropy (Δ*S*^0^)) were obtained through the Van ’t Hoff equation and are shown in Table 3.

Considering the negative values for Δ*G*^0^, it was possible to deduce that the adsorption of tetracycline using the chitosan-based magnetic adsorbent was spontaneous and favorable. In addition, it is notable that a decrease in the values of the Δ*G*^0^ with the increase in temperature hampered the adsorption [1]. Furthermore, negative values for enthalpy and entropy variation revealed an exothermic process, with a decrease in the randomness in the interface solid solution during the adsorption, respectively [21]. Additionally, the Δ*H*^0^ values suggest the occurrence of the chemisorption phenomenon [48].

#### 3.6.7. Comparative Study on the Adsorption Capacity Using Different Adsorbent Materials

The maximum adsorption capacity was compared with previous reports employing different adsorbent materials and conditions. The adsorbent affinity for the adsorbate, pH, temperature, and textural properties are crucial parameters that are involved in the adsorption efficiency [22,49]. Textural properties play an essential role in the adsorption process considering that this phenomenon comprises the mass transfer from the liquid phase to the solid surface. While the specific surface area is associated with the most extensive contact surface and is generally related to high adsorption capacity values, pore size and pore volume exert considerable effects on the shielding and diffusion of molecules [50]. On the other hand, the pH also has a significant influence on the adsorption performance due to the effect on the chemical speciation of adsorbates and the surface charge of adsorbents, which alter their intensity of interactions (attraction or repulsive forces) [14]. At the same time, the temperature combined with other experimental conditions also contributes to the adsorption capacity of the adsorbent toward the adsorbate.

As summarized in Table 4, magnetic chitosan exhibited excellent adsorption capacity compared to other adsorbent materials. In addition, *q_max_* values also were achieved under different experiment conditions. Overall, the specific surface area had a correlation with the adsorption capacity values. Huízar-Felix et al. [51] employed pristine reduced graphene oxide (rGO) and decorated with α-Fe_2_O_3_ nanoparticles. The *q_max_* values decreased with the incorporation of magnetic nanoparticles onto the carbon nanomaterial surface. This significant reduction is caused by the surface effect (the S_BET_ value decreased about 10-fold) and by the surface modification, hindering the interaction between tetracycline and functional groups of the adsorbent. Similar behavior was reported by Ranjbari et al. [52]. Tricaprylmethylammonium chloride-conjugated chitosan hydrogel presented a high value for S_BET_ (0.4727 m^2^ g^−1^) and consequently, a low maximum adsorption capacity. A positive contribution of the specific surface area to the *q_max_* value was previously reported by Zhang et al. [53] when adsorbing tetracycline using a metal-organic framework with S_BET_ of 1408 m^2^ g^−1^. Although the adsorbent showed an excellent performance for the removal of tetracycline from the aqueous medium, studies aiming to determine the efficiency after several adsorption–desorption cycles were not performed.

Erdem and coauthors [54] reported that the chitosan incorporation into the halloysite caused a decrease in the surface area and optimal pH for adsorption. Although the nanocomposite exhibited a lower specific surface area, a higher adsorption capacity was observed. Despite that copper/cobalt ferrite@chitosan has a similar surface area to magnetic chitosan (present study), the *q_max_* value was significantly higher (CS·Fe_3_O_4_, 211.21 mg g^−1^; CuCoFe_2_O_4_, 4.48 mg g^−1^). On the other hand, the thermodynamic parameter revealed that both adsorptions occurred through an exothermic process. In parallel, adsorbents also exhibited similar zero potential of charge, however the functionalization with different magnetic nanoparticles can explain the difference between optimal pH conditions and adsorption efficiency.

In consonance, a study developed by Zhai et al. [16] reported the use of magnetite nanoparticles for TC adsorption. Although the adsorbent had a high surface area (58.8 m^2^ g^−1^), a low value for adsorption capacity was observed (19.6 mg g^−1^). Adsorbents with high surface areas will not necessarily be the most efficient materials since the adsorption intensity may not be related to this parameter, but to the presence of functional groups [12]. Recently, our research group investigated the influence of the amount of magnetite incorporated into the chitosan surface (specific surface area) on methotrexate adsorption. The findings showed that initially, the surface area played an important role in the adsorption efficiency, however, the gradual increase in S_BET_ did not have a direct correlation with the adsorption capacity. Therefore, it was concluded that the adsorption phenomenon was governed not only by cation–π interactions but was also controlled by functional groups (hydrogen interactions) [22]. Corroborating with Huízar-Félix et al. [51], a decrease in the adsorption capacity can be attributed to the occupation of active sites by the magnetic nanoparticles, as well as their binding to the functional groups.

Notwithstanding the studies presenting a promissory alternative to wastewater treatment, a few communications reported desorption studies. This approach is essential, considering the possibility of reducing costs by the application of materials with an excellent capacity for regeneration and reuse. This drawback may be associated with the absence of magnetic behavior, which hampers the easy separation from the aqueous solution.

**Table 4 polymers-14-04854-t004:** Maximum adsorption capacities of different adsorbent materials for TC.

Adsorbents	T (K)	pH	*q_max_* (mg g^−1^)	Reference
Carbon nanotube with 5.9% oxygen content	298	4.0	210.43	[34]
Nanocrystalline cellulose	288	5.0	6.47	[55]
Fe_3_O_4_ nanoparticles	302	7.0	19.6	[16]
Reduced graphene oxide	298	7.0	44.23	[51]
α-Fe_2_O_3_/reduced graphene oxide	298	4.0	18.47	[51]
Halloysite/chitosan nanocomposite	298	8.5	15.6	[54]
Tricaprylmethylammonium chloride-conjugated chitosan hydrogel	318	7.0	22.42	[52]
Mesoporous cage metal-organic framework	323	3.32	442.5	[53]
Copper/cobalt ferrite@chitosan	298	3.5	4.48	[1]
Graphene oxide-functionalized magnetic particles	298	-	39.1	[56]
Rice husk ash	313	5.0	8.37	[57]
Magnetic chitosan	293	7.0	211.21	This work

#### 3.6.8. Regeneration and Reuse

To evaluate the regeneration and reuse ability, the magnetic adsorbent was used in several adsorption–desorption cycles. Batch adsorptions were conducted under the following experimental conditions: initial concentration of TC: 50 mg L^−1^, room temperature, pH = 7.0, and adsorbent dosage: 0.5 g L^−1^. The desorption procedure was carried out using NaOH as the desorbing agent, stirring for 1 h at room temperature. The material was separated by applying a simple magnet, washed with distilled water, and dried in an oven at 50 °C. Figure 14 shows the adsorbent performance in diverse adsorption/desorption cycles. According to the graph, CS·Fe_3_O_4_ presented excellent efficiency after five consecutive cycles. The slight reduction in the percentage removal values can be attributed to incomplete desorption and irreversible tetracycline binding to the active adsorption sites [1,22].

## 4. Conclusions

The chitosan-based magnetic adsorbent was successfully produced by the co-precipitation methods and employed for tetracycline removal from the aqueous solution. The characterization analysis showed that the adsorbent presents a crystalline and mesoporous structure, irregular surface, and ferromagnetic behavior. Through the results, it was possible to confirm that adsorbent performance is directly influenced by the initial concentration of TC, adsorbent dosage, pH, and ionic strength. The highest adsorption capacity was reached at pH 7.0 (211.21 mg g^−1^). The pH changed the chemical equilibrium of the CS·Fe_3_O_4_ and tetracycline. This behavior truly affects the hydrogen bonds, cation–π, and their electrostatic interactions, hampering the binding of the pollutant to the adsorption sites. At the same time, the comparative study related to the effect of magnetite incorporation onto the polymer structure revealed that textural properties also had an important effect on the adsorption efficiency. The equilibrium and kinetic studies were described by Sips and Elovich models, suggesting that the process occurs in heterogeneous surfaces with lateral contributions of TC molecules and predominantly by the chemisorption mechanism. Thermodynamic parameters indicate that adsorption was exothermic, spontaneous, and favorable, with a decrease in the randomness surfaces. The magnetic adsorbent demonstrated efficiency after several cycles of adsorption/desorption. Finally, magnetic chitosan proved to be a promising alternative material for tetracycline removal from the aqueous medium. Additionally, the magnetic behavior (higher saturation magnetization value of 43 emu g^−1^) allowed the easy separation of the adsorbent from the solution and improved the waste management engineering techniques.

## Figures and Tables

**Figure 1 polymers-14-04854-f001:**
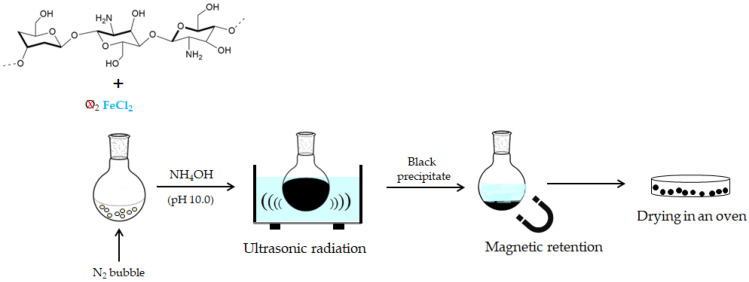
Scheme of synthesis of chitosan-based magnetic adsorbent.

**Figure 2 polymers-14-04854-f002:**
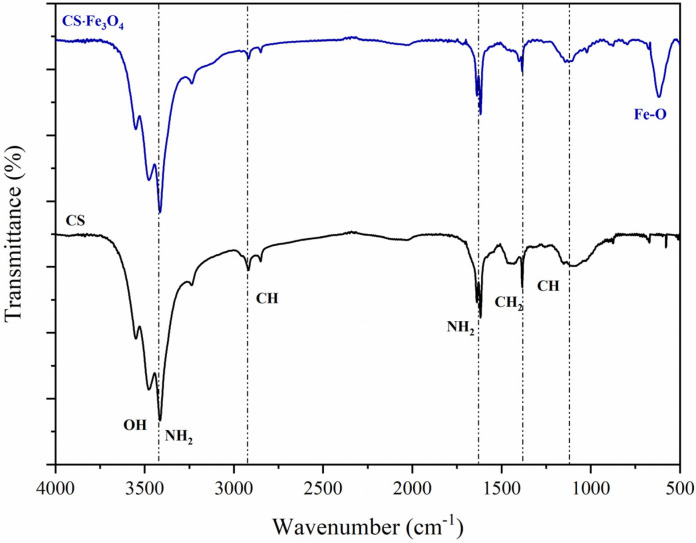
FTIR spectra of chitosan and magnetic chitosan.

**Figure 3 polymers-14-04854-f003:**
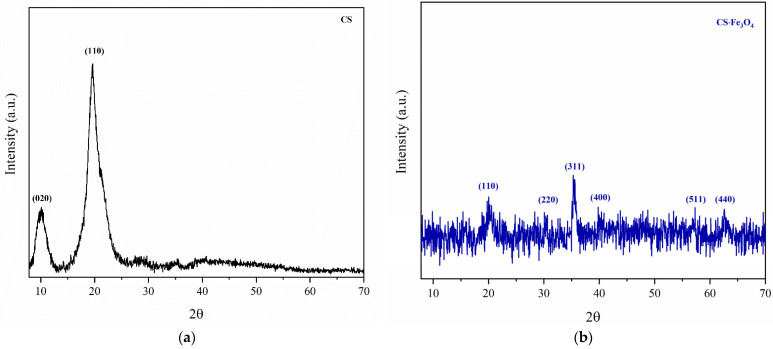
X-ray diffraction: (**a**) chitosan and (**b**) magnetic chitosan.

**Figure 4 polymers-14-04854-f004:**
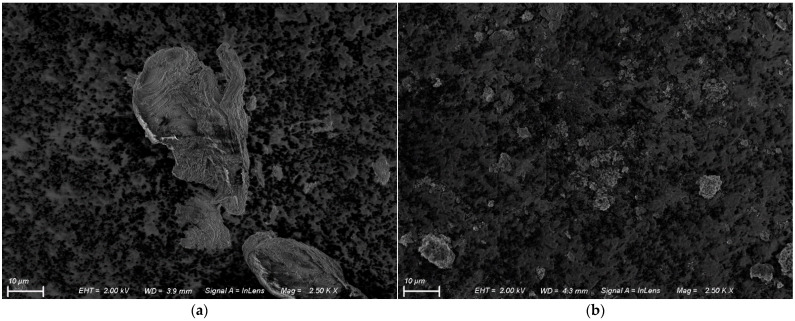
SEM images of biopolymers with 2.5 KX magnification: (**a**) chitosan and (**b**) magnetic chitosan.

**Figure 5 polymers-14-04854-f005:**
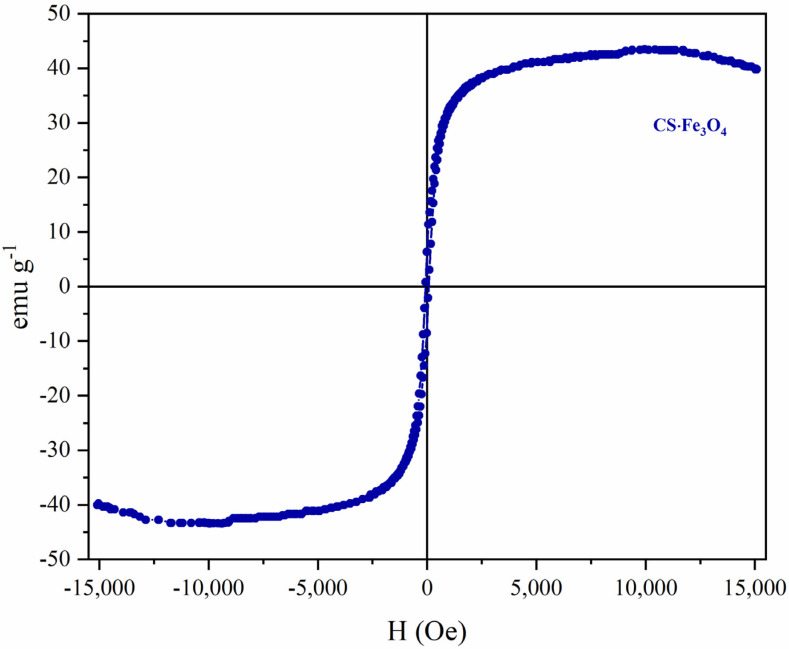
Hysteresis curve of CS·Fe_3_O_4_.

**Figure 6 polymers-14-04854-f006:**
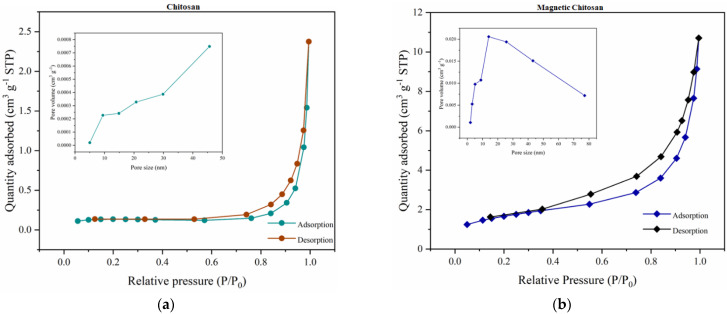
Textural properties of (**a**) CS and (**b**) CS∙Fe_3_O_4_.

**Figure 7 polymers-14-04854-f007:**
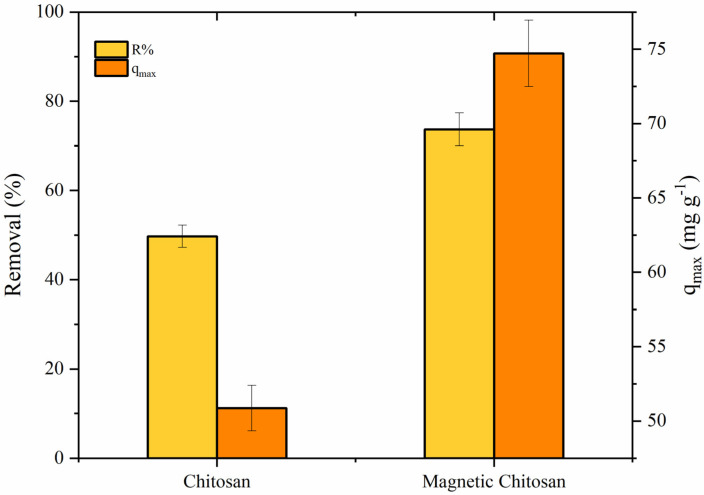
Effect of magnetite incorporation onto chitosan surface. Adsorbent dosage (0.5 g L^−1^), initial concentration of TC (50 mg L^−1^), pH = 7.0, and 293.15 K.

**Figure 8 polymers-14-04854-f008:**
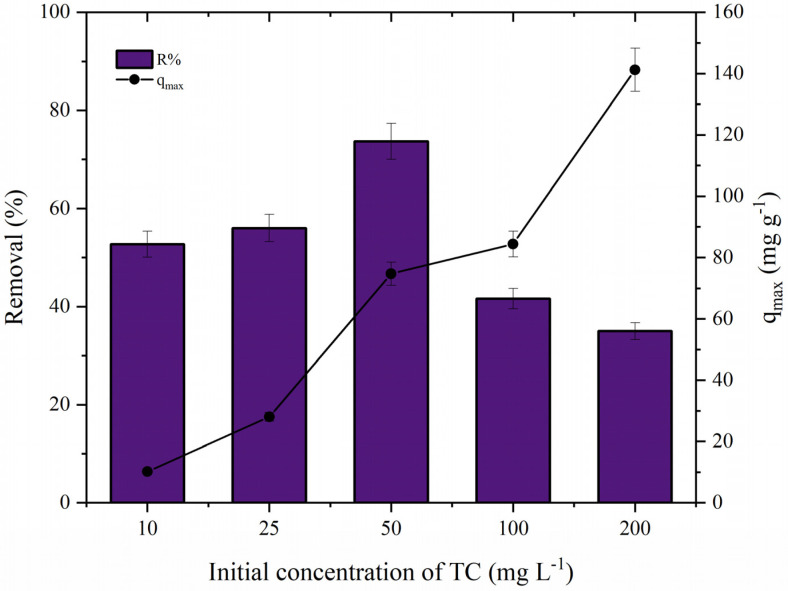
Effect of the initial concentration of TC (CS·Fe_3_O_4_ dosage (0.5 g L^−1^), *C*_0_ = 10–200 mg L^−1^, pH = 7.0, and 293.15 K).

**Figure 9 polymers-14-04854-f009:**
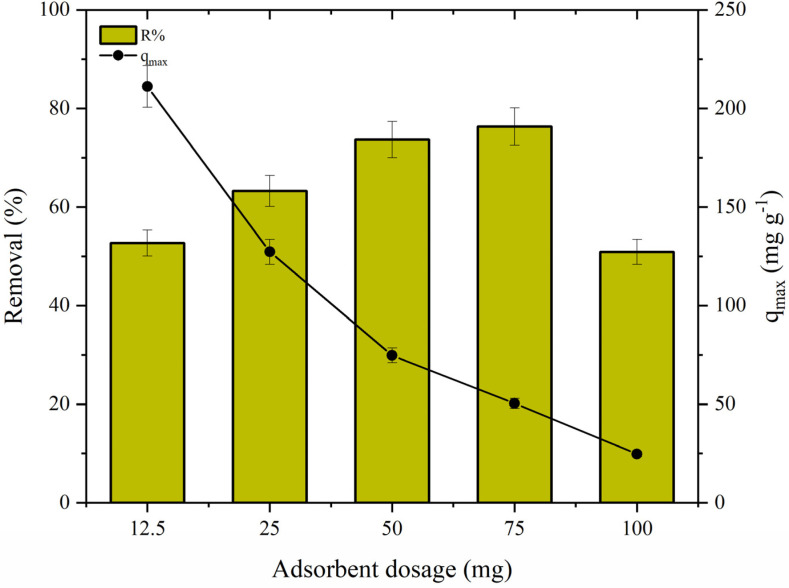
Effect of the adsorbent dosage on TC adsorption (CS·Fe_3_O_4_ dosage (0.125–1.0 g L^−1^), initial concentration of TC (50 mg L^−1^), pH = 7.0, and 293.15 K).

**Figure 10 polymers-14-04854-f010:**
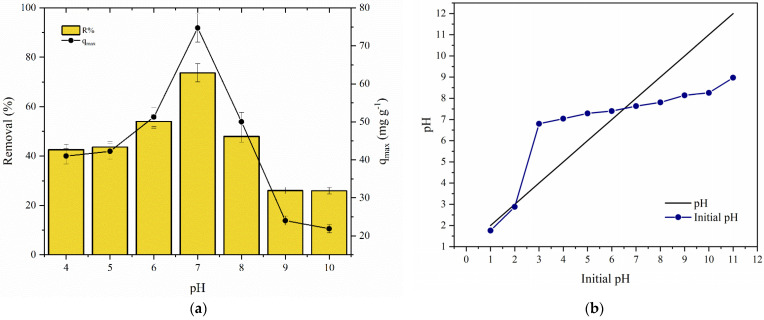
(**a**) Zero point of charge of the CS·Fe_3_O_4_. (**b**) Effect of pH on TC adsorption (*C*_0_ = 50 mg L^−1^, pH = 4–10, adsorbent dosage = 0.5 g L^−1^, V = 100 mL, and 293.15 K).

**Figure 11 polymers-14-04854-f011:**
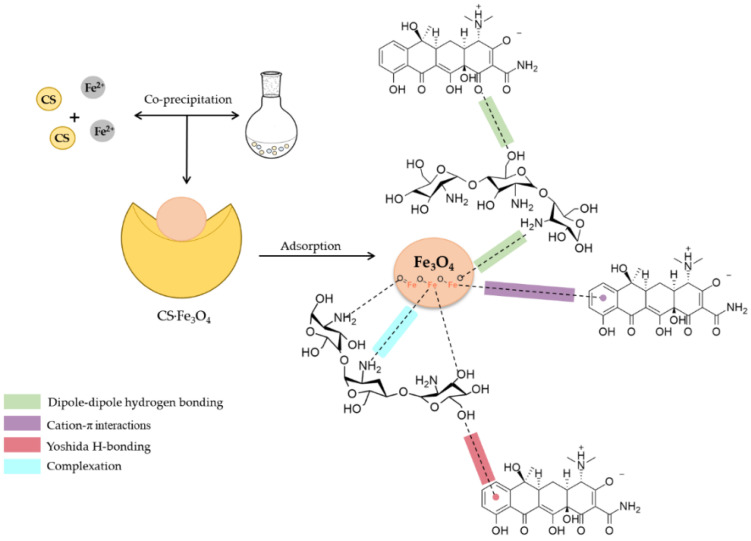
Hypotheses of the tetracycline adsorption mechanism.

**Figure 12 polymers-14-04854-f012:**
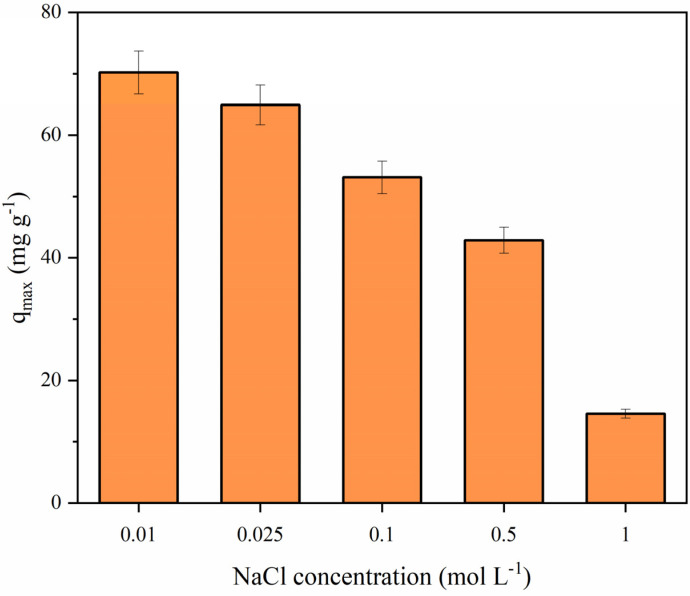
Effect of ionic strength on TC adsorption (*C*_0_ = 50 mg L^−1^, pH = 7.0, adsorbent dosage = 0.5 g L^−1^, NaCl concentration (0.01–1.0 mol L^−1^), V = 100 mL, and 293.15 K).

**Figure 13 polymers-14-04854-f013:**
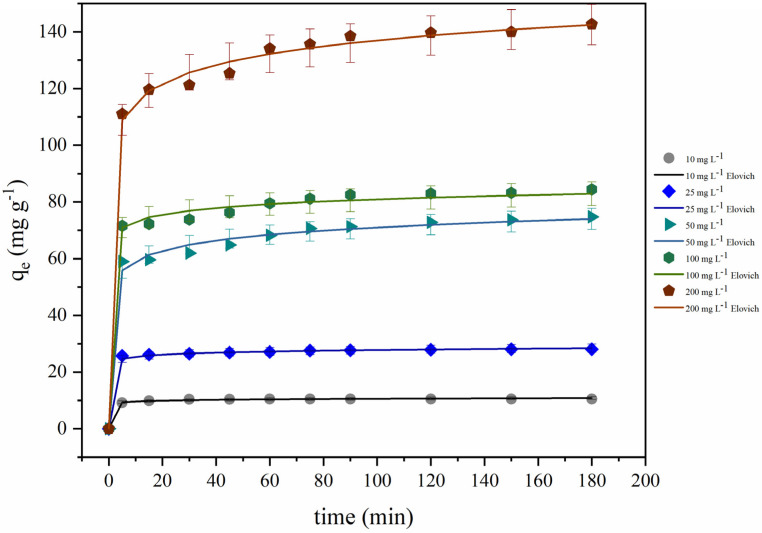
Kinetic curves for TC adsorption on CS·Fe_3_O_4_.

**Figure 14 polymers-14-04854-f014:**
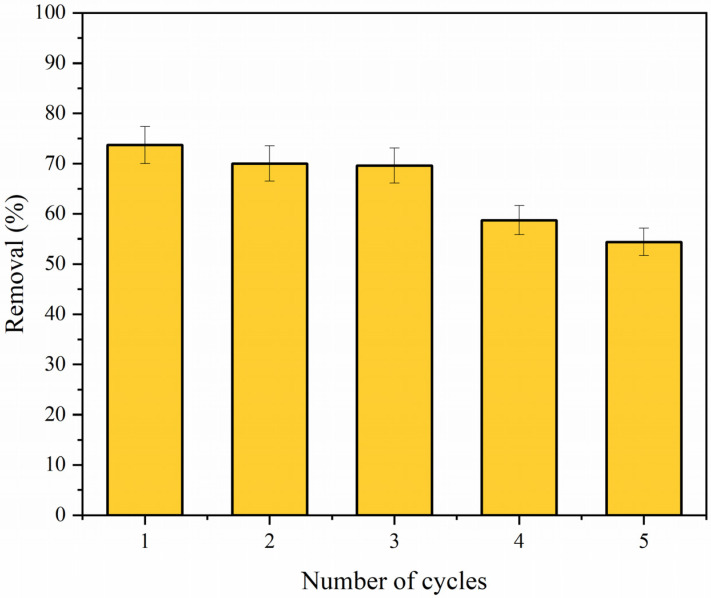
CS·Fe_3_O_4_ performance after several adsorption cycles.

**Table 1 polymers-14-04854-t001:** Kinetic parameters for tetracycline adsorption on CS·Fe_3_O_4_.

Concentration (mg L^−1^)	10	25	50	100	200
**Pseudo-first-order model (PFO)**					
*q*_1_ (mg g^−1^)	10.43	27.31	68.73	79.71	133.17
*k*_1_ (min^−1^)	0.409	0.586	0.378	0.452	0.348
*R* ^2^	0.996	0.994	0.974	0.973	0.964
*R* ^2^ * _adj_ *	0.995	0.992	0.967	0.966	0.955
*ARE* (%)	0.93	1.79	5.47	3.75	4.58
*SSE*	0.02	0.36	6.41	4.24	9.51
**Pseudo-second-order model (PSO)**					
*q*_2_ (mg g^−1^)	10.61	27.61	71.08	81.39	137.54
*k*_2_ (g mg^−1^ min^−1^)	0.112	0.080	0.009	0.013	0.004
*R* ^2^	0.999	0.996	0.987	0.984	0.982
*R* ^2^ * _adj_ *	0.999	0.995	0.983	0.980	0.977
*ARE* (%)	0.39	1.14	4.51	3.22	3.13
*SSE*	0.11	0.20	5.97	8.54	6.65
**Elovich model**					
*α* (mg g^−1^ min^−1^)	1.27	3.91	6.33	11.81	21.23
*β* (g mg^−1^)	2.51	0.958	0.300	0.197	0.106
*R* ^2^	0.997	0.998	0.993	0.996	0.996
*R* ^2^ * _adj_ *	0.996	0.997	0.991	0.995	0.995
*ARE* (%)	0.40	0.85	2.34	1.81	1.35
*SSE*	0.04	0.13	2.72	2.77	5.05

**Table 2 polymers-14-04854-t002:** Equilibrium parameters for TC adsorption on CS·Fe_3_O_4_.

Temperature	20 °C	30 °C	40 °C
**Langmuir**			
*q_max_* (mg g^−1^)	86.81	85.37	92.12
*K_L_* (L mg^−1^)	0.201	0.069	0.045
*R* ^2^	0.977	0.965	0.906
*R* ^2^ * _adj_ *	0.971	0.956	0.882
*ARE* (%)	3.94	4.56	8.84
*SSE*	9.04	9.55	24.97
**Freundlich**			
*K_F_* ((mg g^−1^) (L^−1^)^−1/n^	6.59	3.28	2.51
*n*	43.48	20.22	13.56
*R* ^2^	0.967	0.956	0.892
*R* ^2^ * _adj_ *	0.958	0.945	0.865
*ARE* (%)	4.87	5.21	9.55
*SSE*	13.46	11.89	28.85
**Sips**			
*q_s_* (mg g^−1^)	74.71	65.01	57.03
*K_s_* (L mg^−1^)	0.069	0.064	0.049
*ns*	5.20	3.81	2.74
*R* ^2^	0.998	0.992	0.987
*R* ^2^ * _adj_ *	0.997	0.990	0.983
*ARE* (%)	0.81	3.29	3.02
*SSE*	0.42	4.66	3.84

**Table 3 polymers-14-04854-t003:** Thermodynamic parameters for TC adsorption onto CS·Fe_3_O_4_.

T (K)	Δ*G*^0^ (kJ mol^−1^)	Δ*H*^0^ (kJ mol^−1^)	Δ*S*^0^ (kJ mol^−1^ K^−1^)
293.15	−3.25	−27.29	−0.082
313.15	−1.65		
333.15	−1.19		

## Data Availability

Not applicable.

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
