# Peer review of "Highly Efficient Adsorption of Tetracycline Using Chitosan-Based Magnetic Adsorbent"

_polymers, 2022, doi:10.3390/polym14224854_

Round 1
Reviewer 1 Report
Generally the manuscript is convincing and thoroughly prepared. There are only minor comments.
The following expression in the absract "the adsorption occurs in a heterogeneous system" doesn't make any sense because adsorption by definition is a phenomenon, which occurs on the phase boundaries in heterogeneous systems.
Concerning the results of ΔG calculations: is it correct to keep 3 significant figures in them? If yes, the comments on nonmonotonic changes of the values with temperature variation are necessary.
25% of the references are autocitations, this is certainly excessive.
Author Response
Generally, the manuscript is convincing and thoroughly prepared. There are only minor comments.
Authors: We thank the referee for our paper evaluation, and we consider the comments and suggestions to improve the paper a lot.
Queries are given in black and answers in blue.
The following expression in the abstract "the adsorption occurs in a heterogeneous system" doesn't make any sense because adsorption by definition is a phenomenon, which occurs on the phase boundaries in heterogeneous systems.
A: The expression was rephrased (Line 42). Thank you for the observation.
-Concerning the results of ΔG calculations: is it correct to keep 3 significant figures in them? If yes, the comments on nonmonotonic changes of the values with temperature variation are necessary.
A: Thanks for your observation. In fact, the values data were incorrect in the table. The ΔG tends to decrease with the increase in temperature. The two last values, for 313.15 K and 333.15 K, were changed (Lines 415).
-25% of the references are autocitations, this is certainly excessive.
A: Thank you for the criticism. Considering the requested new table the autocitations automatically were reduced, nevertheless we remove some of our references.
Reviewer 2 Report
The authors should show BET results for chitosan and chitosan based magnetic adsorbent to convince readers that the incorporation of Fe3O4 increased the surface area of the CS.Fe3O4 as stated in line 205-206. The graph for N2-adsorption-desorption showing the type of isotherm (microporous or mesoporous) and the pore size distribution should be included. Line 319, I assume the authors meant concentration of 10 and 25 mg/L not 20mg/L as it reflects on Table 1.
Author Response
Authors: We thank the referee for the positive evaluation of our paper, and we consider all the comments and suggestions to improve the paper.
Queries are given in black and answers in blue.
The authors should show BET results for chitosan and chitosan based magnetic adsorbent to convince readers that the incorporation of Fe3O4 increased the surface area of the CS.Fe3O4 as stated in line 205-206. The graph for N2-adsorption-desorption showing the type of isotherm (microporous or mesoporous) and the pore size distribution should be included.
A: BET results and pore size distribution were included in the manuscript (See main manuscript, Lines 215-230).
Line 319, I assume the authors meant concentration of 10 and 25 mg/L not 20mg/L as it reflects on Table 1.
A: Thank you for your observation. It was corrected.
Reviewer 3 Report
In this work, the author successfully synthesized magnetic chitosan (CS‧Fe3O4) using co-precipitation method and characterized through FTIR, XRD, SEM, and VSM analyses. The as-prepared composites were used as an adsorbent in removal of tetracycline. The findings are of considerable interest and well done. I recommend it to be published after a major revision.
1. The novelty needs to refinement and should be highlighted in the introduction part.
2. The Authors should also proofread their manuscript (some spelling and grammar errors).
3. Maybe the author should compare their results clearly with other reported works, highlighting the advantage and disadvantages of their novel composite.
4. Introduction part, if possible, some important and relative reports about Photocatalysis could helped: https://doi.org/10.1016/j.colsurfa.2021.127753, https://doi.org/10.1016/j.surfin.2022.102006, https://doi.org/10.1016/j.heliyon.2022.e09652.
5. The conclusion is also not targeted to the important aspects described in the manuscript; please rephrase it.
6. The author should better improve the beauty and quality of the figures in the manuscript.
7. In all figures, lack of error bar for the obtained data?
Author Response
In this work, the author successfully synthesized magnetic chitosan (CS‧Fe3O4) using co-precipitation method and characterized through FTIR, XRD, SEM, and VSM analyses. The as-prepared composites were used as an adsorbent in removal of tetracycline. The findings are of considerable interest and well done. I recommend it to be published after a major revision.
Authors: We thank the referee for the considerations for the improvement of our paper.
Queries are given in black and answers in blue.
- The novelty needs to refinement and should be highlighted in the introduction part.
A: The novelty was improved in the introduction part as requested (See main manuscript, Lines 92-97).
- The Authors should also proofread their manuscript (some spelling and grammar errors).
A: Thank you for your observation. The grammar errors were revised and corrected.
- Maybe the author should compare their results clearly with other reported works, highlighting the advantage and disadvantages of their novel composite.
A: A table containing comparative studies were added and discussed (See main manuscript, Lines 415 – 477).
- Introduction part, if possible, some important and relative reports about Photocatalysis could helped: https://doi.org/10.1016/j.colsurfa.2021.127753, https://doi.org/10.1016/j.surfin.2022.102006, https://doi.org/10.1016/j.heliyon.2022.e09652.
A: Thank you for the suggestion. The references were added in the introduction section.
- The conclusion is also not targeted to the important aspects described in the manuscript; please rephrase it.
A: The conclusion was rewritten, improving the important aspects and findings in the current work (See main manuscript, Lines 492 - 513).
- The author should better improve the beauty and quality of the figures in the manuscript.
A: Thank you. It was made.
- In all figures, lack of error bar for the obtained data?
A: The error bar was added in all figures.
Round 2
Reviewer 3 Report
accepted in the present form